

# Accuracy of augmented reality-guided needle placement for pulsed radiofrequency treatment of pudendal neuralgia: a pilot study on a phantom model

Lars L. Boogaard[1,2], Kim Notten[2], Kirsten Kluivers[2],
Selina Van der Wal[3], Thomas J. J. Maal[1] and Luc Verhamme[1]

[1] Radboudumc 3D Lab, Radboud University Medical Centre, Nijmegen, The Netherlands
[2] Department of Obstetrics and Gynaecology, Radboud University Medical Centre, Nijmegen, The Netherlands
[3] Department of Anesthesiology, Pain and Palliative Care, Radboud University Medical Centre, Nijmegen, The Netherlands

Corresponding author
Lars L. Boogaard,
lars.boogaard@radboudumc.nl

## ABSTRACT

**Background:** Pudendal neuralgia (PN) is a chronic neuropathy that causes pain, numbness, and dysfunction in the pelvic region. The current state-of-the-art treatment is pulsed radiofrequency (PRF) in which a needle is supposed to be placed close to the pudendal nerve for neuromodulation. Given the effective range of PRF of 5 mm, the accuracy of needle placement is important. This study aimed to investigate the potential of augmented reality guidance for improving the accuracy of needle placement in pulsed radiofrequency treatment for pudendal neuralgia.

**Methods:** In this pilot study, eight subjects performed needle placements onto an in-house developed phantom model of the pelvis using AR guidance. AR guidance is provided using an in-house developed application on the HoloLens 2. The accuracy of needle placement was calculated based on the virtual 3D models of the needle and targeted phantom nerve, derived from CBCT scans.

**Results:** The median Euclidean distance between the tip of the needle and the target is found to be 4.37 (IQR 5.16) mm, the median lateral distance is 3.25 (IQR 4.62) mm and the median depth distance is 1.94 (IQR 7.07) mm.

**Conclusion:** In this study, the first method is described in which the accuracy of patient-specific needle placement using AR guidance is determined. This method could potentially improve the accuracy of PRF needle placement for pudendal neuralgia, resulting in improved treatment outcomes.

## INTRODUCTION

Pudendal neuralgia (PN) is a chronic and severe neuropathic pain syndrome, that causes continuous pain, numbness and dysfunction in the pelvic region in both men and women

(*Robert et al., 1998*; *Benson & Griffis, 2005*; *Hibner et al., 2010*). Although the International Pudendal Neuropathy Association documented that one in 100,000 in the general population is diagnosed with this syndrome, the actual incidence of PN is unknown. Specialists believe that the incidence is markedly higher than reported and may be comparable to those of asthma (37 per 1,000) and back pain (41 per 1,000) (*Spinosa et al., 2006*; *Hibner et al., 2010*). Frequent causes of PN are pelvic surgery, direct trauma, childbirth, chronic constipation, prolonged sitting, and excessive cycling (*Hibner et al., 2010*; *Ploteau et al., 2017*; *Kaur & Singh, 2021*). Diagnosis of PN is difficult and extensive. To date, the "Nantes Criteria" are generally used for diagnosis (*Pérez-López & Hita-Contreras, 2014*), yet the diagnosis is often made only after many years of symptoms. The poorly understood etiology, pathophysiology and management of this disorder mandate the use of a multidisciplinary approach.

The current state-of-the-art treatment of pudendal neuralgia often involves a trial-and-error approach to obtain the best results for any individual patient. Novel therapies are emerging such as pulsed radiofrequency (PRF) therapy, which has been successfully employed as a treatment modality for several neuropathic pain syndromes, including pudendal neuralgia (*Masala et al., 2014*; *Kalava, Pribish & Wiegand, 2017*; *Fang et al., 2018*; *Frank et al., 2019*; *Krijnen et al., 2021*). When performing PRF treatment, a needle is supposed to be placed close to the targeted nerve, resulting in treatment of the nerve although the exact working mechanism of PRF remains unclear. The most frequently mentioned mechanism of action is that the short pulses of PRF lead to an alteration in synaptic transmission, resulting in neuromodulation (*Cahana et al., 2006*; *Chua, Vissers & Sluijter, 2011*; *Sam et al., 2021*). The tip of the PRF needle generates a high energy field that decreases exponentially as the distance from the tip increases (*Cosman & Cosman, 2005*; *Chua, Vissers & Sluijter, 2011*). It is stated that PRF must be applied within 5 mm of the nerve to be effective (*Sluijter & van Kleef, 1998*; *Deniz, Bakal & Inangil, 2016*). Those findings highlight the importance of accurate needle placement.

Even though PRF seems a promising technique in the treatment of patients with PN, there are some limitations. An important limitation is the fact that the pudendal nerve cannot be visualized directly during the procedure. Hence, landmark-based (*i.e.*, ischial spine and internal pudendal artery) methods are currently used to localize the pudendal nerve, among which the ultrasound (US)-guided transgluteal method is the most common in outpatient clinics (*Ploteau et al., 2017*). Another guidance technique used is Computed Tomography (CT)-guidance. However, the accuracy of PRF needle placement using these landmark-based methods is still uncertain. Although imaging techniques can identify landmarks in a clinical setting, the size and tract of the pudendal nerve vary between individuals, making accurate localization of the pudendal nerve challenging (*Montoya et al., 2011*; *Ploteau et al., 2017*; *Leslie et al., 2021*).

Due to limited data from clinical studies, pulsed radiofrequency is not yet recommended as an evidence-based treatment for PN (*Levesque et al., 2022*). The few studies investigating PRF treatment for PN show promising results. Effect sizes of around 90% are reported after 2 to 6 months of follow-up (*Fang et al., 2018*; *Krijnen et al., 2021*; *Wang & Song, 2022*). However, there are disparities in the definition of effective treatment.

*Wang & Song (2022)* distinguished between at least a 25% reduction in VAS score (88.9% effective) and at least a 50% reduction in VAS score (54.1% effective) at 12 weeks of follow-up. In our clinic, we found that 70% of the patients had a clinically important pain reduction after PRF treatment (two points on the numeric rating scale). In addition, variations in effectiveness in consecutive treatments are observed. Given the importance of accurate PRF needle placement, this raises the question of whether PRF treatment for PN is currently performed accurately continuously.

Augmented reality (AR) is a novel navigational tool allowing the user to see three-dimensional (3D) virtual objects overlaying real-world objects. In literature, various studies have successfully implemented AR technology in medical visualization (*Vávra et al., 2017*; *Carl et al., 2019*; *Pérez-Pachón et al., 2020*). This could enhance the intervention by improving the visualization of the pudendal nerve by overlaying its tract onto the patient's surface. Thereby, the individual anatomy can be visualized, which may result in better treatment results (patient-specific treatment). We aim to explore the potential benefits of AR in PN treatment and thereby fill the knowledge gap concerning accurate needle placement. In this study, a simulated pudendal nerve will be projected onto a phantom model of the pelvis based on landmark alignment. The accuracy of needle placement using AR guidance will be examined. We hypothesized that the needle can be placed within the PRF range of the pudendal nerve using AR guidance of the transgluteal approach.

## MATERIALS AND METHODS

### Anthropometric phantom model

A sawbone anthropomorphic phantom of the pelvis was used to represent the anatomy of the pelvic region. The phantom contains the anatomic structure of the pudendal nerve pathway. Starting from the formation of the main trunk and ending at the ischial spine where it branches. A copper wire of 75 mm in length and 2 mm in diameter was used to simulate the pudendal nerve (*O'Bichere, Green & Phillips, 2000*). The contour of the gluteal muscles was constructed with paper. The puncture area was covered with an 87 mm × 57 mm × 21 mm sponge.

A total of six validated glass markers (DentalMark 1.0 mm; The Suremark Company, Mesa, AZ, USA) were included in the model representing the left and right ischial tuberosity, the left and right midpoint lateral border sacrum and the left and right greater trochanter (Fig. 1). These structures serve as landmarks for the alignment of the AR projection.

### 3D computer models and segmentation

The phantom model was scanned using a cone-beam computer tomography (CBCT) scanner (OP 3D Vision V17; KaVo Dental, Biberach, Germany). The scan protocol had the following settings: voxels 0.3 mm isotropic, peak voltage 120 kV, and X-ray tube current 5 mA.

The Digital Imaging and Communications in Medicine (DICOM) images obtained from the CBCT scan were 3D reconstructed using the software Mimics Medical (version

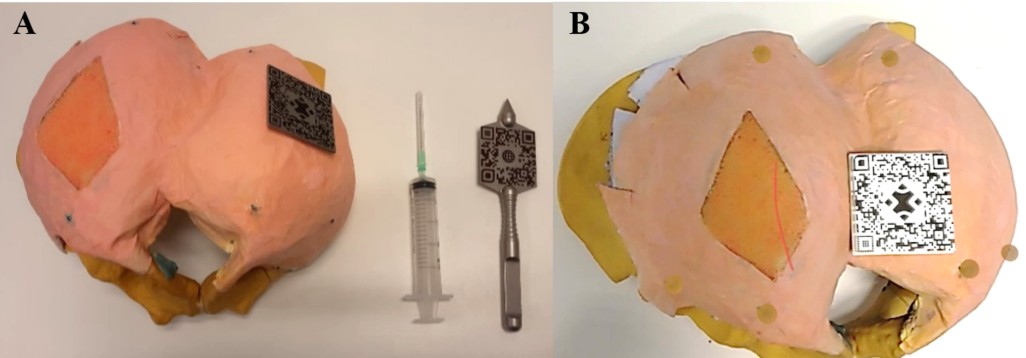

**Figure 1** **Pictures of the in-house develop model of the pelvis the equipment.** (A) Dorsal view of the phantom model with the gluteal area covered in pink, and the section of injection in orange, corresponding to the transgluteal approach. The glass markers are colored green, representing the left and right ischial tuberosity, left and right midpoint lateral border sacrum and left and right greater trochanter. QR codes for the alignment of the AR application on both the phantom model and an in-house made stainless steel pointer. The needle is attached to a syringe for puncture. (B) Holographic overlay of the 3D segmented pudendal nerve (red) onto the phantom model. Landmark markers are highlighted in brown.                                              

28.0; Materialise NV, Leuven, Belgium). The bone, nerves, and six glass markers were segmented with different threshold values.

## Augmented reality application

An AR application was developed using the game engine Unity (Unity Technologies, 2019.3.11) and the marker tracking method from Vuforia AR SDK (PTC Inc., Boston, MA, USA). The application was deployed in the Microsoft HoloLens 2 (Microsoft Corporation, Redmond, WA, USA). The workflow is as follows: (1) image acquisition and segmentation of the relevant anatomical structures, (2) importing the relevant anatomy to an in-house-developed AR application, (3) real-time tracking of the object, and (4) registration and visualization of the holographic anatomy.

Real-time tracking of the phantom model was accomplished through the utilization of two 50 mm × 50 mm stainless steel Quick Response (QR) codes places strategically on the phantom model and a custom-made stainless steel pointer. When the AR application was initiated, pinpointing of landmarks was required using this pointer. The HoloLens two registered the positions of both the phantom model and the pointer as reference.

The alignment between the virtual and real-world environment was achieved automatically using the Procrustes algorithm. This mathematically algorithm optimized translation and rotation to minimize differences between the corresponding points in the dataset, ensuring precise overlay and visualization of the digital model (anatomy).

## Accuracy study

To examine the potential of AR-guided PRF needle placement, nine observers conducted the pilot experiment. All subjects had no prior experience with the PRF treatment nor the use of AR, to exclude the effect of experience. Since this study includes a phantom model,

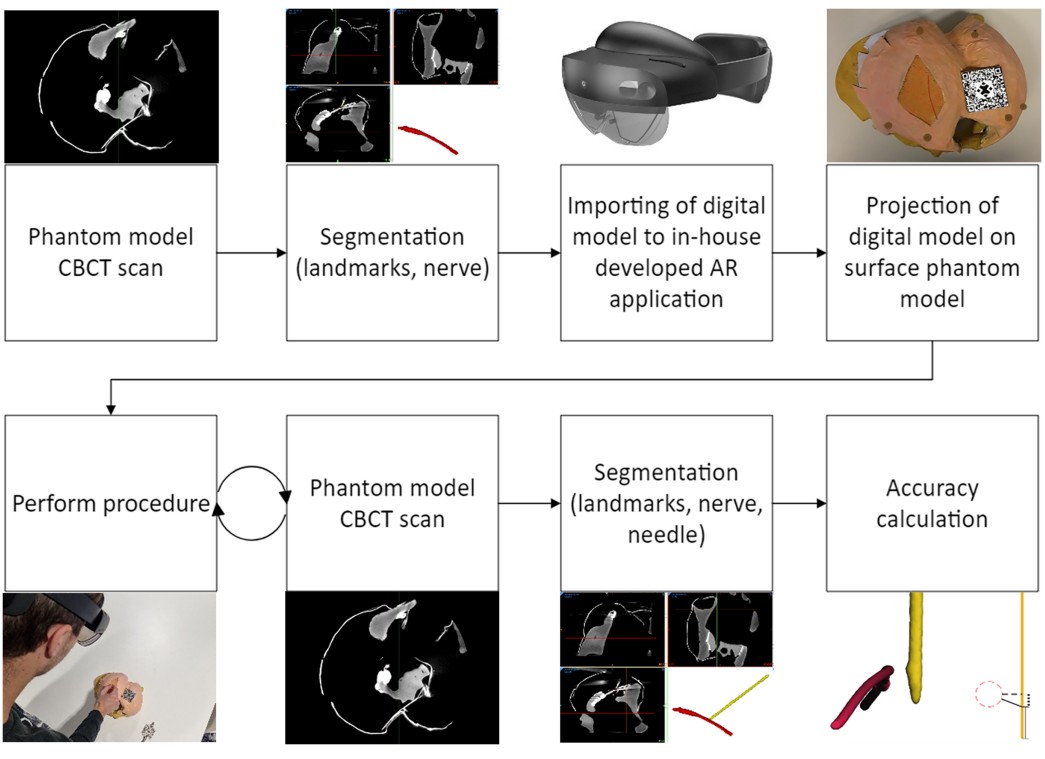

**Figure 2 Flowchart of the experimental design.**

this study did not meet the criteria for human subject research and was thus exempt from the Ethical Committee review.

At the start of the experiments, the HoloLens 2 was calibrated for every subject individually. The standard eye calibration embedded in the HoloLens 2 was used. During this process, you'll look at a set of targets (gems). Focusing on the gems allows the HoloLens 2 to learn about the eye position to render the virtual word. Thereafter, the subjects were asked to bring both QR codes in their sight. Once the subjects verbally confirmed that both QR codes did light up, the subjects were instructed to pinpoint the six landmarks in the model using the pointer tool: the left and right ischial tuberosity, left and right midpoint lateral border sacrum and left and right greater trochanter. A ball shape appeared over the landmarks once it was pinpointed, to provide feedback to the subject regarding the alignment. The subjects were able to redo their pinpointing. After all landmarks were pinpointed, the simulated nerve was projected onto the phantom model as described in the previous section (Fig. 1). The subjects were instructed to focus on the nerve for a minimum of 1 and a maximum of 5 min to ensure familiarity with the gestures of the HoloLens 2.

Thereafter, the subjects were asked to puncture the phantom model and place the tip of the needle as close as possible to the nerve tract. A specific target on the nerve tract was not specified, which corresponds to the current procedure performed in our clinic. After each trial, the needle was left in the phantom model and a CBCT scan was made, using the scan protocol described above. The data were stored for analysis. A flowchart of the experimental design is shown in Fig. 2.

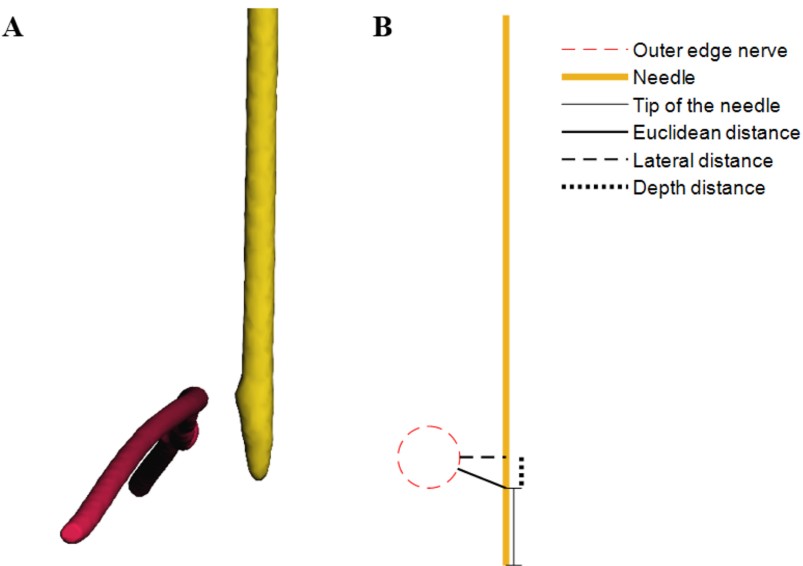

**Figure 3** (A) Virtual 3D models discplaying the nerve in red and needle in yellow, derived from CBCT scans of the model. (B) Schematic overview illustrating the definitions of accuracy.

The nerve and needle were segmented separately using threshold masking in the Mimics Medical software (version 28.0; Materialise NV, Leuven, Belgium), with the same Hounsfield-Unit (HU) threshold applied for every scan. The resulting nerve and needle were exported as separate Standard Tessellation Language (STL) files for further analysis.

The needle and nerve placement assessment was conducted using 3DMedX® software (version 1.2.29.3; 3D Lab Radboudumc, Nijmegen, The Netherlands). To match the nerves of each trial in 3D space, an iterative closest point (ICP) algorithm within the 3DMedX program was utilized. The resulting transformation matrix of the matched nerve was then applied to the needle, which aligned all structures to the simulated nerve. The tip of the needle was defined as all data points in the outermost 5.00 mm. These points correspond to the active tip of the most commonly used PRF needles. The maximum Euclidean distance between consecutive 3D coordinates was 0.01 mm. All calculations were performed using MATLAB software (version R2020b; The Mathworks Inc., Natwick, MA, USA).

### Primary outcomes

The primary outcome of this study was the accuracy of needle placement, which is subdivided into (1) the minimal Euclidean distance between the tip of the needle and the nerve, (2) the lateral distance between the needle and the nerve and (3) the distance between the tip of the needle and the lateral distance points on the needle (*i.e.* depth distance). The most relevant parameter is the minimal Euclidean distance between the tip of the needle and the nerve. A schematic overview of the definitions is shown in Fig. 3.

### Statistical analyses

All datasets were assessed for normality using the Shapiro-Wilk test. Data will be presented by the median and interquartile range (IQR). To analyze if the accuracy is within the PRF

**Table 1 The results of the accuracy study.**

|  | Median (mm) | IQR (Q1 − Q3) (mm) | % <5.00 mm | *p* |
|---|---|---|---|---|
| Euclidean distance | 4.37 | 5.16 (2.01 − 7.17) | 55 | 0.382 |
| Lateral distance | 3.25 | 3.62 (0.94 − 4.56) | 83 | <0.001 |
| Depth distance | 1.94 | 7.07 (0.10 − 7.17) | 70 | 0.157 |

**Note:**
Q1 = 25% percentile. Q3 = 75% percentile. % <5.00 mm = percentage of trials within the range of 5 mm of the target.
*p* = *p*-value of the one-sample Wilcoxon signed-rank test with 5 mm as the hypothesized median (target value).

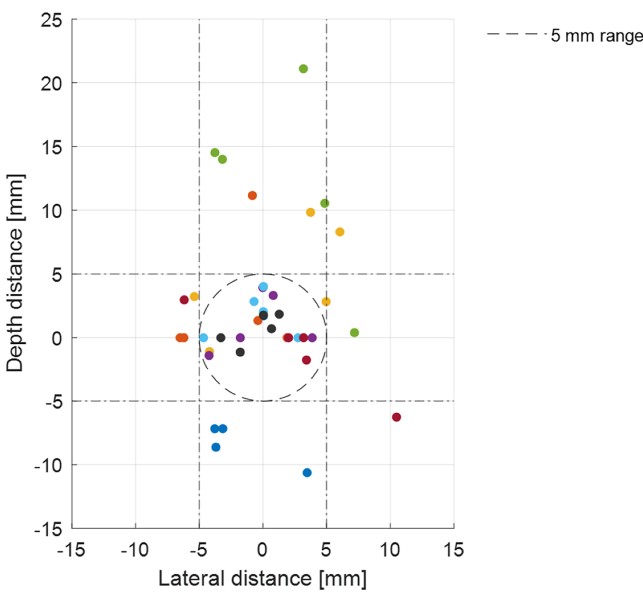

**Figure 4 Accuracy of the tip of the needle from all subjects.** The origin represents the midline of the nerve (target). The x-axis corresponds to lateral distance, in which a positive integer means the needle is placed to the right of the target from the view of the subject. The y-axis corresponds to the depth of the needle placement, in which a negative integer means the needle is placed deeper than the target. The effective range of 5 mm is shown by the dashed circle and dashed lines.

effective range of 5.00 mm, one-sample t-tests were performed to test statistical significance for normally distributed samples, and the one-sample Wilcoxon signed-rank test for non-normally distributed. For this study, *p*-values of <0.05 were considered statistically significant. All statistical analyses were performed using IBM SPSS for Windows/MAC (version 27.0; IBM, Armonk, NY, USA).

## RESULTS

All trials of eight subjects were included for analysis, leading to a total of forty trials. One subject was not able to experiment, because the AR projection was visibly not correctly aligned even after recalibration, which was likely the result of the glasses of the subject. Shapiro-Wilk tests showed that the data is not normally distributed (*p* = 0.021, 0.036 and <0.001 for respectively the Euclidean distance, the lateral distance and the depth distance).

The median Euclidean distance was 4.37 (IQR 5.16) mm, the median lateral distance was 3.25 (IQR 3.62) mm and the median depth distance was 1.94 (IQR 7.07) mm. The tip of the needle was placed within the range of 5.00 mm in 55% of the trials ($p = 0.382$). In 83% of the trials, the lateral distance was within the range ($p < 0.001$) and in 70% the depth of the needle was within 5.00 mm of the nerve ($p = 0.157$). The interquartile ranges show that the greatest variance is found in the depth distance. The results of the accuracy study are shown in Table 1 and Fig. 4.

## DISCUSSION

This is a report on a pilot study in which AR is used as a guidance tool for PRF needle placement on a pelvic phantom model. We found that the tip of the needle (Euclidean distance) can be placed at a median of 4.37 mm of the nerve. The median lateral distance between the needle and the nerve was 3.25 mm. The largest IQR is found to be in the depth direction.

The pudendal nerve is difficult to localize, mainly due to the absence of bony landmarks and anatomical variations, making puncturing the pudendal nerve accurately challenging. Accurate needle placement could lead to better PRF treatment outcomes and improve the quality of life of various patients. AR guidance has the advantage of being a personalized navigation method based on patient-specific anatomy. In literature, various studies show promising results regarding accurate needle placement using AR guidance (*Rau et al., 2021*; *Demerath et al., 2022*, among others). The findings of this study support these results and is the first to describe an AR guidance system targeting the pudendal nerve.

There is a belief that the accuracy of PRF needle placement through AR guidance will outperform the accuracy through US guidance and therefore improved treatment outcomes (*i.e.*, pain relief) and reduced treatment variability is expected. A cadaver study can be conducted to compare the accuracy of conventional (US-guided) and AR-guided PRF. In addition, the clinical effect (*i.e.*, pain reduction) of PRF treatment of both methods should be compared to determine the added value of AR guidance.

The determined Euclidean distance in this study includes errors in the system, the visualization and the user. These errors were estimated in a previous study by *Meulstee et al. (2019)* who found an average Euclidean distance of 2.3 mm with a maximum error of 3.6 mm when using the HoloLens2 in a preclinical phantom study. The AR-visualization error was estimated to be around 1.5 mm (*Meulstee et al., 2019*).

In our study, a large IQR was observed for the depth distance variable, indicating high variability and uncertainty in this direction, and thus a relatively high associated error. This is most likely due to the absence of depth feedback during the procedure. To improve precision and accuracy, visual aids could be implemented to guide the subject to correct deviations and stop at the appropriate depth. *Demerath et al. (2022)* included the planned trajectory in their method of AR guidance of drain placement on a phantom model. The authors found a median Euclidean distance of 3.0 mm (IQR 1.7 mm) between the drain placement and the target using AR guidance, which was statistically significantly lower than using the conventional (freehand) method (*Demerath et al., 2022*), implying that adding visual aids increase the accuracy compared to our proposed method.

On the other hand, adding feedback induces more complexity to the method and introduces additional errors of unknown magnitude. A study by *Van den Bosch et al. (2022)* showed that real-time position feedback using AR guidance resulted in an accuracy range of 1.9 ± 0.9 mm to 7.5 ± 5.3 mm. The large difference between participants and the large standard deviation, as well as the failure to convincingly improve the accuracy compared to our findings, implies that the additional CT guidance used may be of limited advantage to our proposed method. As a result, a thorough evaluation of the particular method, including of feedback given a certain surgical setup would be necessary. It has been chosen to limit the complexity of the procedure, to determine the baseline accuracy of AR-guidance of PRF treatment of the pudendal nerve and thereby propose a first method. Moreover, it must be acknowledged that the experiment in this study is conducted on a phantom model of the pelvic. This model does not fully represent the complexity of the pelvic floor anatomy.

The main future development lies in the visualization of the pudendal nerve before the treatment, which is necessary for patient-specific navigation. Although there have been promising advancements in MRI techniques, such as MR Neurography (MRN) (*Chhabra, Madhuranthakam & Andreisek, 2018*) and diffusion tensor imaging (DTI) (*van der Jagt et al., 2012*; *Lemos et al., 2021*), there is currently no published literature on the visualization of the pudendal nerve and integration in a navigation system (*e.g.*, AR-guidance).

Additionally, while our study focused on the transgluteal approach, other approaches such as the transvaginal and transperineal approaches are also used to access the pudendal nerve and could potentially benefit from AR guidance. If this method shows promising results, it would of course not be limited to pudendal neuralgia alone, but the indication could be extended to other PRF treatments for neuralgia in the pelvic floor or other areas.

## CONCLUSIONS

In the present article, we describe a method to test AR-guided PRF for pudendal neuralgia. Although the tip of the needle was not always within an optimal 5 mm margin from the nerve, the lateral distance shows promising results in terms of accuracy. This method has the potential to enhance the accuracy and thereby personalize needle placement for PRF treatment, conceivably leading to improved outcomes for patients with pudendal neuralgia. Further research is required to identify the optimal technique for AR-guided PRF and to evaluate its effectiveness.

## ACKNOWLEDGEMENTS

The authors thank S. Orozco Carvallo for developing the phantom model and J. Duits for developing the AR application.

### Funding

The authors received no funding for this work.

## Competing Interests

The authors declare that they have no competing interests.

## Author Contributions

- Lars L. Boogaard conceived and designed the experiments, performed the experiments, analyzed the data, prepared figures and/or tables, authored or reviewed drafts of the article, and approved the final draft.
- Kim Notten conceived and designed the experiments, analyzed the data, authored or reviewed drafts of the article, and approved the final draft.
- Kirsten Kluivers analyzed the data, authored or reviewed drafts of the article, and approved the final draft.
- Selina Van der Wal analyzed the data, authored or reviewed drafts of the article, and approved the final draft.
- Thomas J. J. Maal conceived and designed the experiments, analyzed the data, authored or reviewed drafts of the article, and approved the final draft.
- Luc Verhamme conceived and designed the experiments, analyzed the data, authored or reviewed drafts of the article, and approved the final draft.

## Data Availability

The models and raw data are available in the Supplemental Files.

## Supplemental Information

Supplemental information for this article can be found online at http://dx.doi.org/10.7717/peerj.17127#supplemental-information.

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
