# Peer review of "Accuracy of augmented reality-guided needle placement for pulsed radiofrequency treatment of pudendal neuralgia: a pilot study on a phantom model"

_PeerJ, doi:10.7717/peerj.17127_

## Round 0.1 · original submission · Major Revisions

- Although the basic reports appear to be correct, there are some details described by the reviewers that need to be addressed.
- Please detail the response for each reviewer and in case of disagreement justify your response adequately.

Reviewer 1 ·

Basic reporting

The paper generally adheres to English standards, with clear and unambiguous language. However, it lacks a statement regarding the sharing of raw data, which is an essential aspect of research transparency. it is recommended to detail out whether the data are available for validation or replication.

Experimental design

1. The research question is NOT well-defined. It must be relevant and meaningful so that it may aim to explore the potential benefits of augmented reality (AR) in pudendal neuralgia (PN) treatment.
2. The study does mention the significance of filling a knowledge gap in the introduction, but this could be more explicitly stated.
3. The investigation appears to be rigorous, with adherence to ethical standards and the use of a phantom model for experimentation, which provides a controlled environment. However, the paper lacks certain technical details regarding the AR system and tracking equipment, which are crucial for replicability. A more comprehensive description of these aspects is needed.
4. It would be good to have a graphic representation of complete experimental design from gap toward validation.
5. It is must to describe the training provided to the subjects in more detail. How was their proficiency with AR assessed before the actual experiment?

Validity of the findings

1. The paper does not assess the impact and novelty of the findings. It is essential to discuss how these findings contribute to the existing literature and whether they represent a novel approach to treating PN. The absence of this discussion limits the overall significance of the study.
2. Data availability is not addressed in the paper neither it is been explain. Providing a statement about data availability is standard practice in research and helps ensure transparency and replication. It is recommended to give a graphical representation
3. Conclusions are reasonably well-stated and linked to the original research question. However, it is recommended to discuss the clinical implications of using AR in PN treatment more explicitly.
4. The presentation of results is clear, but the text lacks interpretation. The authors should discuss the clinical significance of their findings in more detail.
5. It would be helpful to include statistical measures of dispersion (e.g., standard deviation) along with medians and IQRs to provide a more comprehensive view of data distribution.
6. Consider presenting the results in a table or figure to enhance readability.

Additional comments

1. The abstract provides a good overview, but it lacks a clear statement of the study's significance. It is recommended to give a brief statement on the potential clinical impact of the findings would be beneficial.
2. The research objectives are somewhat vague. It recommended to clearly state the specific aims and hypotheses of the study in the Introduction section.
3. The description of the phantom model and the materials used is clear and informative.However, more details about the AR application and the tracking system, including technical specifications should be in added.
4. It is recommended to add a flowchart or diagram illustrating the experimental procedure which could aid in understanding the study design
5. Ethical considerations are briefly mentioned, but it's important to explain why the study was exempt from ethical review. Was it because it involved a phantom model? Clarification is needed.
6. It is must to describe the training provided to the subjects in more detail. How was their proficiency with AR assessed before the actual experiment?
7. The presentation of results is clear, but the text lacks interpretation. The authors should discuss the clinical significance of their findings in more detail.
8. It would be helpful to include statistical measures of dispersion (e.g., standard deviation) along with medians and IQRs to provide a more comprehensive view of data distribution.
9. Consider presenting the results in a table or figure to enhance readability.
10. There's a mention of errors in the system, but it would be valuable to quantify and discuss the sources of error and their potential impact on the study's outcomes.
11. Please address the limitations of the study more comprehensively. For instance, the use of a phantom model may not fully represent the complexities of human anatomy, and this should be acknowledged.
12. Suggest areas for future research, such as clinical trials to validate the effectiveness of AR-guided needle placement in actual patients.

·

Basic reporting

In relation to basic reporting there is no need to make any modifications in my opinion.

Experimental design

In relation to the experimental design, the reconstruction of the experimental model seems to me to be very accurate. I think it provides enough information to be able to understand each step of the process.
Although I can assume from reading the article that CBCT scanner images are used, I think it would be much clearer for readers unfamiliar with the process of transferring images from diagnostic devices to augmented reality devices, if the original CBCT image (axial view where we can see part of the target zone), the segmented image and the projection of that image onto the experimental model were shown.
As a suggestion you can see the flow-chart shown in the article on echo-guided vascular puncture assisted by augmented reality (Morillas Pérez J, Mechó Meca S, Caballero Galindo G, Miguel Pérez-Llano J. Validation of the effectiveness of augmented reality-assisted vascular puncture: An experimental model. The Journal of Vascular Access. 2023;0(0))

Validity of the findings

The results support the possibility that augmented reality-guided puncture improves the placement of the needle tip close to the pudendal nerve to ensure that the radiofrequency treatment is delivered to the most appropriate location.
The way the position of the needle tip is shown in figure 3 allows a very clear visualisation of the results of the process.

Additional comments

This is a very promising study on the use of augmented reality-guided needle placement for pulsed radiofrequency treatment of pudendal neuralgia. I found the article very valuable in stimulating and promoting further research into new technologies in this field. Congratulations on the work.

From lines 214 to 217, you explain that the biggest deviation from the target is in the depth accuracy. It is proposed to use some kind of guide to correct this deviation. Have you considered measuring the depth on the CBCT scan and marking the needle so that, when it is inserted to the mark, the tip is at the desired depth?

---

## Round 0.2 · accepted · Accept

The authors have properly addressed all of the reviewers' comments.

·

Basic reporting

There are no further suggestions

Experimental design

There are no further suggestions

Validity of the findings

There are no further suggestions

Additional comments

Having read the authors' comments in reference to the reviewers' suggestions and the actions they have taken, I have no further suggestions that would significantly improve the excellent work that has been done.
My recommendation is that the article would be ready for publication.